# CFD-Modified Potential Simulation on Seakeeping Performance of a Barge

Seol Nam [1], Jong-Chun Park [1,*], Jun-Bum Park [2] and Hyeon-Kyu Yoon [3]

1   Department of Naval Architecture and Ocean Engineering, Pusan National University, Busan 46241, Korea
2   Division of Navigation Science, Korea Maritime and Ocean University, Busan 49112, Korea
3   Department of Naval Architecture and Ocean Engineering, Changwon National University,
    Changwon 51140, Korea
*   Correspondence: jcpark@pnu.edu

**Abstract:** This study proposes a computational fluid dynamics (CFD)-modified potential (CMP) model. This hybrid model uses linear potential theory with a corrected damping ratio obtained from CFD simulations to analyze the seakeeping performance of a small vessel. According to the analysis procedure of the proposed model, a motion analysis, including the prediction of the roll and pitch damping ratio of a small barge, was conducted; to verify reliability; the results were compared to those of an experiment performed in a physical tank. The relative errors in the experiment for peak amplitude in the roll motion response amplitude operators (RAOs) using the CMP model were relatively small, whereas those obtained from only the potential analysis were large errors in all three conventionally used roll-damping ratios. In addition, the computational time consumed by the CMP model was longer than that consumed by the potential theory but faster than the full CFD simulation for all wave conditions. Subsequently, based on the motion analysis results, the seakeeping performance was evaluated in a real sea environment, and the results on the single significant amplitude (SSA) were discussed through comparison with the results of the potential analysis and experiment.

**Keywords:** CFD-modified potential model; seakeeping performance; barge; damping ratio; motion RAO

## 1. Introduction

According to data from the Korea Maritime Safety Tribunal (2020) on the status of maritime accidents in the coastal areas of the Korean Peninsula from 2016 to 2020 by ship types, small- and medium-sized passenger ships and fishing vessels accounted for over a third of all maritime accidents. The stability of small ships is at a higher risk of being affected by a wave environment, not only by the roll motion but also by other planar motions, including the pitch motion [1]. Excessive ship motion causes an increase in fatigue among people on board and reduces their work capacity; excessive ship motion can also damage the hull owing to repeated motion [2]. Therefore, to reduce accidents in small ships and ensure safe operation by examining dynamic stability, including roll and pitch, according to the operating state and sea state, it is necessary to quantitatively evaluate the seakeeping performance at the design stage.

In a study on the seakeeping performance of small ships, Tello et al. [3] analyzed the hydrodynamic performance of nine measures of seakeeping performance for 11 types of small fishing vessels based on theoretical equations. Nurhasanah et al. [4] calculated the RAOs of roll, pitch, and heave motions for the operation of small fishing vessels using theoretical equations to evaluate seakeeping performance. However, applying linear theory to model the actual physical phenomenon may be an oversimplification [5]. The model may not represent the phenomenon perfectly, which may degrade the accuracy of its results. Therefore, in many recent studies on analyzing seakeeping performance, model tests in a

physical wave basin have been employed. Kim et al. [6] evaluated seakeeping performance influenced by the hull form of small high-speed vessels. Niklas and Karczewski [7] evaluated hull resistance and seakeeping performance using different hull forms through model tests for small passenger ships, and Seo et al. [8] performed a similar evaluation for small high-speed wave-piercing vessels. However, except for a few studies, in most dynamic stability analyses of small vessels, experiment-based verification was not performed owing to economic and time limitations. Instead, most studies relied on numerical analyses using linear theory and subjective experience values.

To address this issue and obtain high-accuracy results, recently published studies have included numerical results obtained using computational fluid dynamics (CFD) simulations. Niklas and Pruszko [9] solved the motion RAO and evaluated seakeeping performance according to the hull form of small boats via CFD, based on a finite volume method. Lin and Lin [10] evaluated the seakeeping performance at various sailing speeds of small high-speed vessels. Furthermore, Fitriadhy et al. [11] employed CFD to evaluate the seakeeping performance of small naval vessels. However, CFD can incur high computational costs if a large number of calculations are required owing to various frequency and wave-direction conditions; thus, it is a somewhat inefficient method.

Hence, most numerical studies have attempted to evaluate seakeeping performance using the potential flow analysis technique, which has high computational speeds under various frequency and wave direction conditions. Yi et al. [12] conducted motion and seakeeping performance analyses of potential flows based on finite water depth in the frequency domain. The potential theory, which is widely used in seakeeping performance analysis of large-scale vessels, is suitable for analyzing the heave and pitch motion of the hull with a relatively small amplitude. However, there are limitations to predicting roll motion because of the nonlinear effects of roll damping caused by fluid viscosity not considered in potential theory. To fix the problem, previous studies introduced a concept of artificial viscous damping in the potential theory, but it is not easy to determine the appropriate artificial viscous damping due to the high nonlinearity of the roll motion [13]. Consequently, in most cases, roll motion is predicted using empirical formulas for the roll damping coefficient obtained from experiments on other ships. However, the calculation results inevitably have lower accuracy than model tests or CFD simulations. Moreover, in the case of small vessels, as the ship speed increases, the acceleration increases not only because of the roll motion but also because of the pitch motion. This may impact bottom safety because of slamming, and the roll motion is considerably affected by the combined pitch-roll motion [14]. However, there are no accurate recommendations for the damping ratio required to predict pitch motion, and further research on this may be necessary.

Kim et al. [15] proposed an alternative simulation model that maximizes the advantages of potential analysis and CFD simulations and establishes an evaluation procedure for the seakeeping performance of small vessels. The motion RAO solved using potential analysis is calculated when a vessel is operating at a constant speed in waves and corrects the damping ratio of the final motion through the RAO obtained by performing a CFD simulation at the frequency corresponding to the maximum response (i.e., the maximum value of the ship response spectrum calculated by multiplying the square of the RAO by the wave spectrum). However, the maximum value of the ship response spectrum may vary depending on the wave spectrum at the target sea site, and it may be difficult to exclude some ambiguities, such as the occurrence of multiple peaks in the ship response spectrum. Consequently, the results may differ if the damping ratio of the final motion is derived based on the maximum value of the ship response spectrum. Moreover, the reliability of the results was not verified experimentally.

This study makes a unique contribution to the literature by proposing an enhanced simulation model, the CFD-modified potential (CMP) model, which is based on the simulation model suggested by [15]. In this model, the damping ratio at the peak point of the motion RAO was employed for the correction instead of that at the peak of the ship response spectrum. In Section 2, the computational procedures of the CMP model are

briefly introduced and the corresponding computations are described for each process. Furthermore, the verification process of the reliability of the enhanced model through a comparison with an experiment is presented. In Section 3, the seakeeping performance is evaluated using the motion RAOs obtained by applying the CMP model. Subsequently, the results are compared with those of the experiments and potential theory alone, and further discussion is provided regarding the applicability of the CMP model.

## 2. Methods: Numerical Simulation

### 2.1. Introduction of CFD-Modified Potential (CMP) Model

Figure 1 summarizes the computational procedure of motion analysis using the proposed CMP model.

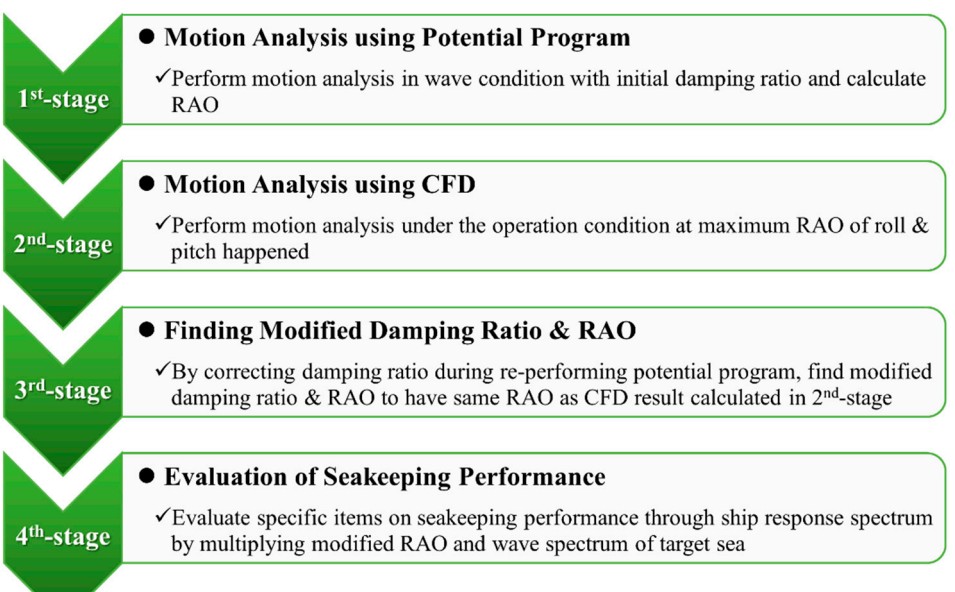

**Figure 1.** The computational procedure of CMP model.

In the first stage, a 6-Degrees of Freedom (DOF) motion analysis of a ship is performed using a potential-based program, considering only wave conditions, and the RAO is calculated. From this step, information on the natural frequency for each motion mode can be obtained. In general, damping ratios are required to perform motion analysis with a potential program. They are generally obtained through a decay test in still water using a model test or CFD simulations.

In the second stage, CFD simulations are performed under the operating conditions of a specific wave period (i.e., the natural period) and wave direction from which the peak RAO was previously calculated. Here, the operating condition is one that considers the ship's speed in addition to the wave environmental condition if the ship is advancing. Whereas roll motion has the most dominant value for large ships, small vessels experience excessive motion from pitch and roll, which significantly impacts both stability and safety. Therefore, CFD motion analysis was conducted for both the roll and pitch RAOs under the conditions for which the maximum value occurred for each.

In the third stage, to determine the appropriate damping ratio, motion analysis is iteratively performed by correcting the damping ratio applied to the potential analysis until the peak value of the RAO equals that calculated from the CFD results in the second stage. This modified damping ratio is different from the initial damping ratio in still water used in the first stage, and the ship speed, if any, can also be considered.

In the fourth stage, the seakeeping performance was evaluated using the ship response spectrum, calculated by multiplying the modified RAO and wave spectrum of the target

sea site. The specific evaluation items for the seakeeping performance are presented in Appendix D.

## 2.2. Target Model and Sea Environmental Condition

As shown in Figure 2, the target model is a barge with a wide and flat bottom. The hull shape is symmetrical about the xz-plane, but two skegs are installed only at the stern. The principal dimensions of the barge at full scale are listed in Table 1.

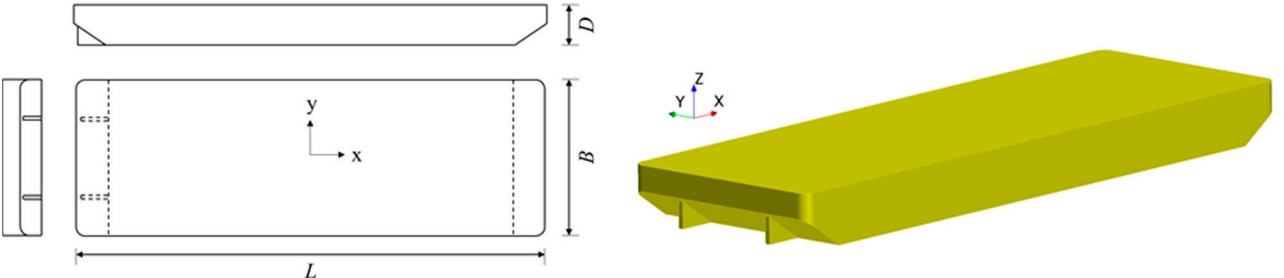

**Figure 2.** Barge model.

**Table 1.** Principal dimensions of the barge.

| Principal Dimension | Full-Scale |
|---|---|
| Length between perpendicular, Lpp (m) | 39.2 |
| Breadth, B (m) | 13.0 |
| Depth, D (m) | 3.3 |
| Draft, T (m) | 1.523 |
| Displacement weight, W (kgf) | 735,000 |
| Longitudinal center of gravity, LCG from AP (m) | 19.6 |
| Vertical center of gravity, VCG from BL (m) | 2.713 |
| Metacentric height, GMt (m) | 7.794 |
| Speed, U (m/s) | 3.086 |

For the seakeeping performance analysis, the Shinan sea area, located off the southwest coast of South Korea, where the barge operates, was selected as the target sea site, as shown in Figure 3. The red "x" is the location of the buoy. A total of 32,160 data samples (one-hour intervals) on wave height and mean zero-crossing wave period were collected from an observation buoy in the Shinan sea area over five years from 2015 to 2019. The data was sourced from the "Meteorological Data Open Portal" provided by the Korea Meteorological Administration [16]. The collected data were classified by significant wave height at 0.1 m intervals and mean zero-crossing wave period at 0.5 s intervals, as shown in the wave scatter table in Figure 4. Here, the number of wave occurrences in the table is the number of wave data values less than the significant wave height and the mean zero-crossing wave period, which are the classification criteria. It is classified based on the World Meteorological Organization's sea state code standard and as a result [17], the occurrence of waves corresponding to sea state 2 (Hs = 0.1–0.5 m) comprised 93% of the total. Although the evaluation of seakeeping performance is typically conducted in a high sea state, which includes extreme conditions, in this study, we analyzed seakeeping performance in the sea state that most frequently occurred at the target sea site, with a focus on introducing the procedure for the proposed method and demonstrating the verification process. Accordingly, Hs = 0.5 m and Tz = 3.5 s, which exhibited the largest wave energy in sea state 2, were selected as the representative wave conditions prevalent at the target sea site, and the seakeeping performance was evaluated based on the operational conditions of the barge.

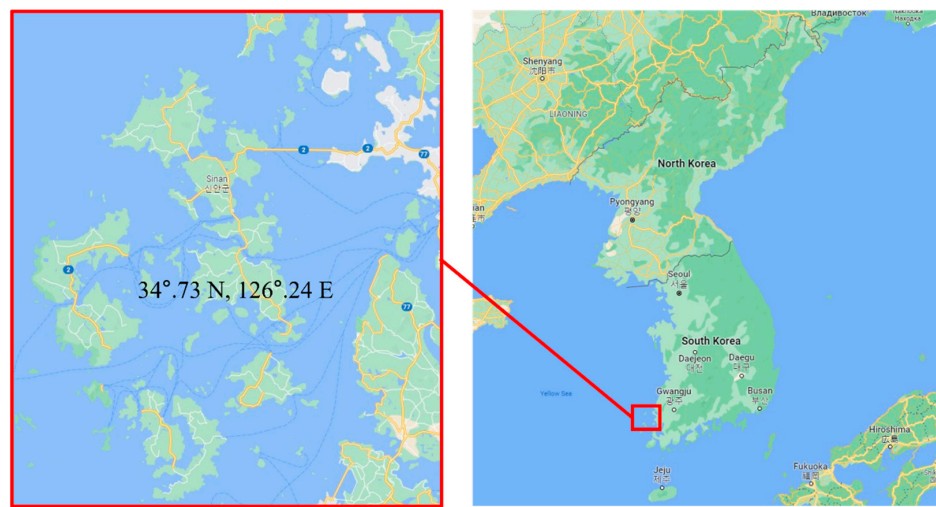

**Figure 3.** Shinan sea data collection point.

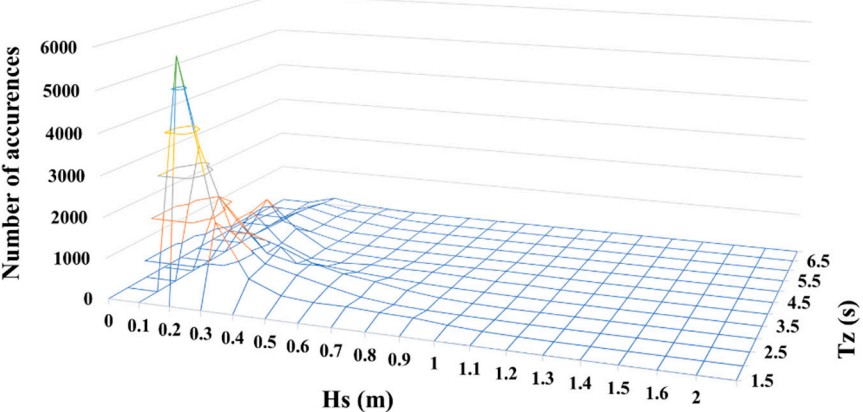

**Figure 4.** Number of wave occurrences.

The Texel–Marsen–Arsloe (TMA) spectrum [18] was used as the wave spectrum, and Suh et al. [19] reported an example of statistically analyzing the appropriate spectrum by examining the wave height and period data of a coast in South Korea. Please refer to Appendix A for a brief introduction to the TMA spectrum and the equation used to calculate it. Figure 5 shows the calculated TMA spectrum of an actual sea site.

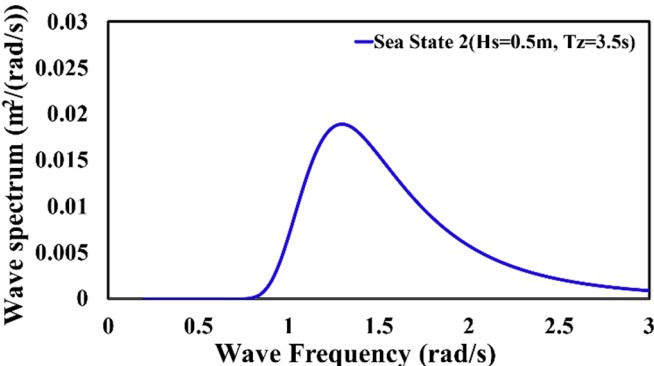

**Figure 5.** TMA spectrum in Shinan.

We collected 17,522 samples of current data generated in 2019 at 30-min intervals. Figure 6 shows the occurrence of the current in terms of direction and speed and red line shows the number of occurrences. As shown in the figure, the current at the target sea

site occurs in an approximately straight line. To minimize hull roll, which has the most significant impact on the hull's stability, the head sea of the barge was assumed to be 157.5°, which is the direction in which the current occurs most frequently. Currents mainly occurred between the speeds of 0.1 m/s and 1.1 m/s, and large currents of 1.2 m/s or higher did not occur on average.

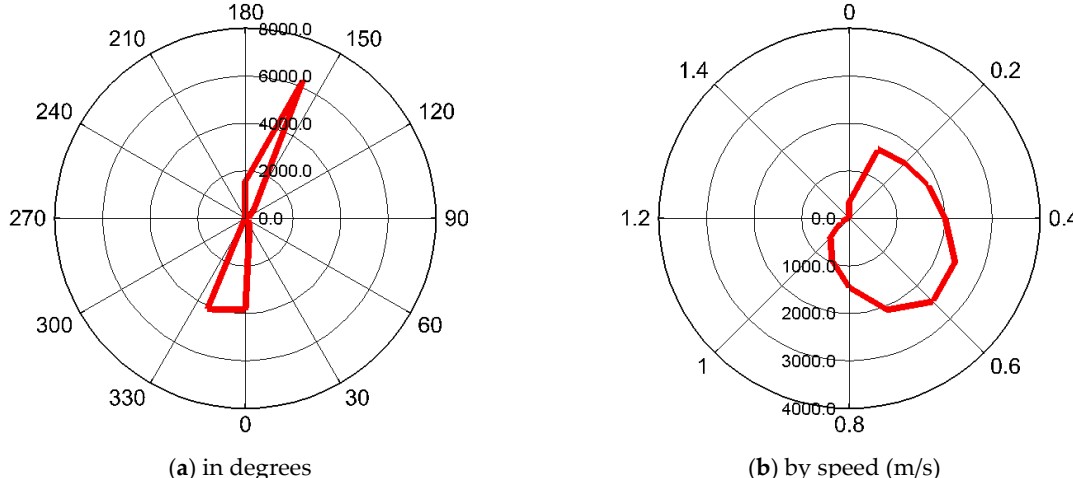

(**a**) in degrees  (**b**) by speed (m/s)

**Figure 6.** Number of current occurrences according to direction.

*2.3. Numerical Simulation Based on CMP Model*

2.3.1. Motion Analysis Using Potential Program (1st-Stage)

In the first stage of the CMP model, motion analysis was conducted on the barge under wave conditions by applying the initial damping ratio using WAVELOAD-FD [20], which is a potential-based analysis program of Lloyd's Register. For the analysis, approximately 30,000 subsurface mesh cells were generated, as shown in Figure 7, and the conditions of the motion analysis are listed in Table 2. To calculate the motion RAO according to the wave direction, the directions were set at 30° intervals from 180° to 0°, and the frequency was set at 0.05 rad/s-intervals from 0.2 rad/s to 3.0 rad/s, based on the ABS Guidance Notes (2003) [21]. In this study, a water depth of 20 m was applied, which was the average depth at the target sea site.

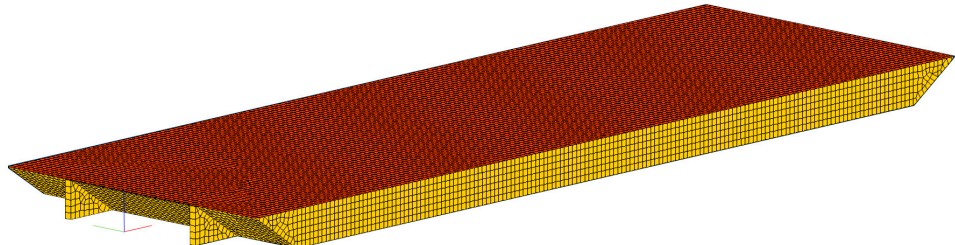

**Figure 7.** Mesh configuration for the motion analysis using a potential program.

**Table 2.** Potential program motion analysis conditions.

| Parameters | Full-Scale |
| --- | --- |
| Wave heading (°) | 0–180, interval 30 |
| Wave frequency (rad/s) | 0.2–3.0, interval 0.05 |
| Water depth (m) | 20 |
| Ship speed (m/s) | 3.086 |
| Damping ratio, $\zeta$ (-) | 0.05, 0.10, 0.15 |

Hulls with a length greater than the width are significantly affected by viscosity during roll motion, which impacts hull stability. When using the potential program, the roll-damping ratio ($\zeta$) must be applied to reflect the fluid viscous force in still water, which, however, cannot be reflected owing to the nature of potential-based analysis. The difference in the RAO motion was first confirmed in this study using three arbitrary damping ratios from 0.05 to 0.15, which is the range generally applied based on experience. Figure 8 shows the RAO for the roll motion when different damping ratios are applied in the 90° wave direction. As observed, the degree of roll motion varies significantly with the damping ratio. Moreover, because pitch-damping ratios are not typically applied in motion analysis, the pitch was not considered in the initial motion analysis. However, in the CMP model procedure, the analysis results in the first stage, which apply the initial damping ratios for the roll and pitch motion, can be regarded as the primary results, which must be corrected by applying modified damping ratios in the third stage.

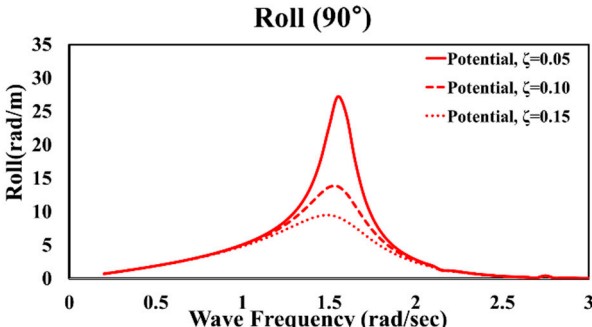

**Figure 8.** Comparison of roll RAO according to damping ratios in wave direction of 90°.

Figure 9 shows the initial RAO of the roll and pitch in each wave direction with an initial roll-damping ratio of 0.05. For roll, the largest motion occurs in a wave direction of 90°, and the corresponding natural frequency is 1.5 rad/s. For pitch, the largest RAO was observed at a frequency of 1.0 rad/s and a wave direction of 180°.

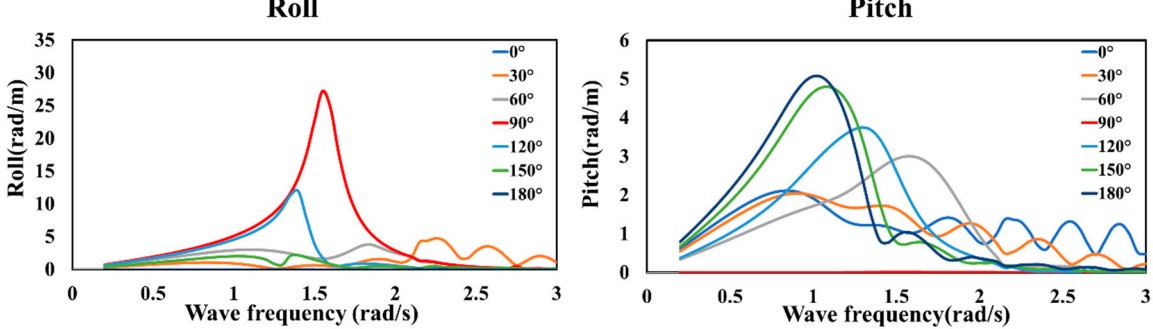

**Figure 9.** Motion RAO of roll and pitch according to wave directions ($\zeta$ = 0.05).

### 2.3.2. Motion Analysis Using CFD (2nd-Stage)

In the second stage, motion analysis is performed using CFD under operation conditions where the peak RAO occurs with reference to the results of the first stage. In this case, it is of paramount importance to guarantee sufficiently reliable and accurate CFD simulation results that can replace experiments. Therefore, we focused on ensuring the reliability of the CFD results through grid-convergence tests.

The commercial program STAR-CCM+15.02 (Siemens PLM Software, Plano, TX, USA) was used for the Reynolds-averaged Navier–Stokes (RaNS)-based CFD simulations. The numerical wave tank (NWT) concept used in the simulation is illustrated in Figure 10. An overset mesh was employed inside the NWT to analyze the motion of the object; further, at the interface between the generated overset region and the NWT, physical quantities were

exchanged in a least-square manner. The NWT used to reproduce the wave environment is fixed; however, the overset region around the hull expresses six-degrees-of-freedom (6-DOF) motion with respect to the hull's center of gravity according to the hull's dynamic behavior. The dynamic fluid body interaction (DFBI) model was used to express the 6-DOF hull behavior. This 6-DOF motion solver can calculate the new position of the ship by solving the equation of motion, which involves calculating the fluid force and moment acting on the ship.

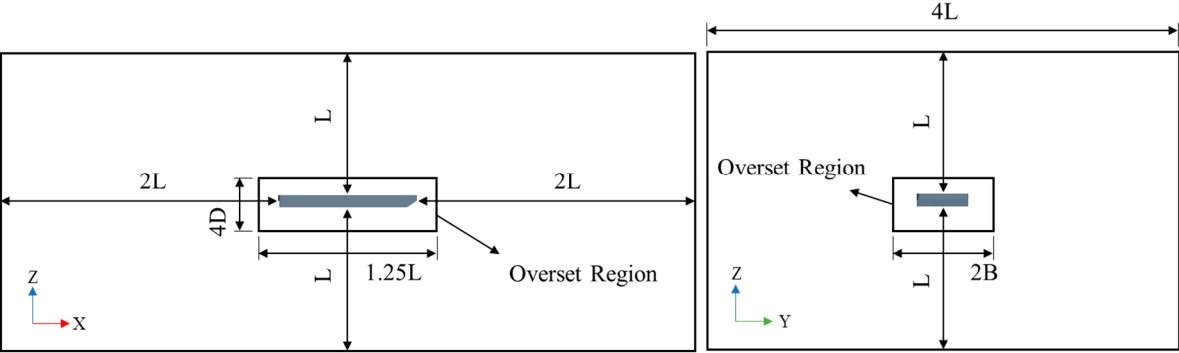

**Figure 10.** Dimension of numerical wave tank for CFD simulation.

For wave generation, as shown in Figure 11, the velocity condition of the airy wave theory was imposed on all velocity inlet interfaces, and the wave forcing technique in STAR-CCM+ was introduced to the wave absorber used to suppress reflected waves [22]. According to Kim et al. [23], the wave forcing technique combines two methods—one that reduces the wave amplitude using the wave-damping coefficient in the wave-absorption area [24–26] and the other that forces the reflected waves by mathematically harmonizing the wave information obtained from the linear wave theory in a given area between the waves inside the flow field and the given incident waves [27,28]. In this study, one wavelength (1.0λ) was set for all the velocity inlet interfaces to suppress the reflected waves as much as possible. For the simulation, the physical model was configured as shown in Table 3 with reference to [29]. A 2nd-order implicit unsteady solver for time and a 2nd-order upwind/central scheme solver for space were used as simulation solvers. The SST (Shear Stress Transport) $k - \omega$ model was used as the turbulence model. The simulation was performed for 5 s with a time interval of 0.001 s.

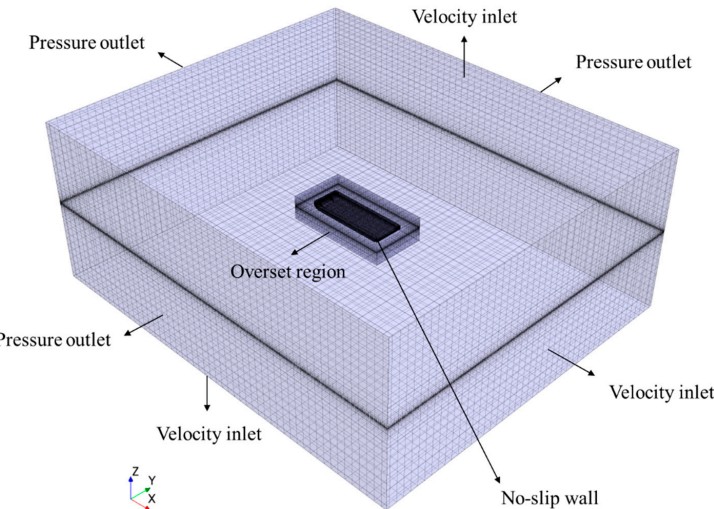

**Figure 11.** Boundary condition and mesh configuration of CFD simulations.

**Table 3.** Physics models for CFD motion simulation.

| Modeling Items | Physics Model |
|---|---|
| Time | Implicit unsteady |
| Turbulence model | SST k-omega |
| Wall function | Low y+ wall treatment |
| | Volume–of–Fluid (VOF) |
| Free-surface | Stokes first order VOF waves |
| | VOF wave forcing |

For grid generation, a surface remesher and trimmed cell mesher were used, and a prism layer mesher was additionally used to implement the fluid viscosity effects close to the moving hull. Eight prism layers were used around the hull, and y+ was configured to be 4 or less.

As it is essential to guarantee the reliability of the CFD results, the grid convergence index (GCI) test, as defined by Roache [30], was performed to evaluate the convergence due to the grid density. Appendix B summarizes the GCI analysis process. This study conducted the grid convergence test for five grid sizes. Refinement factor $r$ was selected to be 1.33, and the base sizes were adjusted. Table 4 and Figure 12 show the total number of mesh cells for each case and their corresponding RAO values. According to the results, the RAO tended to converge to a constant value as the mesh became denser from mesh cases 1 to 5. To evaluate the results quantitatively, three mesh levels were combined to calculate the GCI as shown in Table 5. Here, the recommended value of 1.25 was used as the safety factor. $GCI_2^{coarse}$ shows the values of r, p, and GCI for four mesh levels: (1, 2, 3), (2, 3, 4), (3, 4, 5), and (1, 3, 5). For example, $GCI_1^{fine} = F_s|E_1|$ (1, 2, 3) was calculated from the results of Case 5 (finest) and Case 4 (intermediate), and $GCI_2^{Coarse} = F_s|E_2|$ (1, 2, 3) was calculated using the results of Case 4 (intermediate) and Case 3 (coarsest). Here, $E_1^{fine} = \frac{\varepsilon}{1-r^p}$ and $E_2^{coarse} = \frac{r^p \varepsilon}{1-r^p}$ as shown in the Appendix B. The ratio is also shown to examine whether the result when h = 0 converges within an asymptotic range. According to the GCI analysis, both $GCI_1^{fine}$ and $GCI_2^{coarse}$ exhibited the lowest mesh levels (3, 4, and 5). Hence, the mesh model used in Case 4 was selected, and motion analysis was conducted using approximately 6.5 million mesh cells.

**Table 4.** Mesh convergence test conditions and results.

| Mesh Case | Base Size | Mesh Cells | RAO |
|---|---|---|---|
| 1 | 0.35 | 1 M | 10.37 |
| 2 | 0.27 | 1.6 M | 9.97 |
| 3 | 0.20 | 3.2 M | 9.65 |
| 4 | 0.15 | 6.6 M | 9.52 |
| 5 | 0.11 | 14 M | 9.51 |

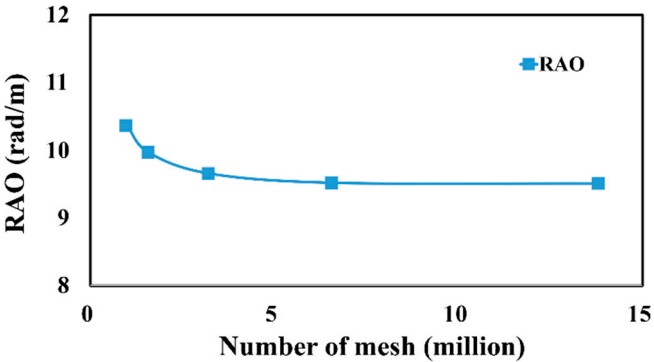

**Figure 12.** Mesh convergence test result.

**Table 5.** Grid convergence index.

| Parameter | Mesh Level 123 | Mesh Level 234 | Mesh Level 345 | Mesh Level 135 |
|---|---|---|---|---|
| r | 1.3300 | 1.3300 | 1.3300 | 1.7689 |
| p | 0.8621 | 2.8681 | 9.1031 | 2.7520 |
| $GCI_1^{fine}$ | 14.5229 | 1.4318 | 0.0109 | 0.5124 |
| $GCI_2^{coarse}$ | 17.9881 | 3.1977 | 0.1460 | 2.4242 |
| ratio | 0.7820 | 0.4413 | 0.0746 | 0.2081 |

The motion analysis using CFD for the vessel in operation was conducted under the wave direction and wave period conditions for which the RAOs for roll and pitch were maximized in the initial potential analysis results, as shown in Figure 9, under the conditions listed in Table 6. The current was considered by applying the advancing speed of the ship, which was satisfied by taking the inlet velocity of the fluid as 3.086 m/s in the STAR-CCM+.

**Table 6.** Wave conditions for which the maximum RAO occurred in the results of a potential program.

| Motion | Advancing Ship Speed (m/s) | Wave Height (m) | Wave Frequency (rad/s) | Wave Direction (°) |
|---|---|---|---|---|
| Roll | | 0.5474 | 1.5 | 90 |
| Pitch | 3.086 | 1.2317 | 1.0 | 180 |

### 2.3.3. Finding Modified Damping Ratio and RAO (3rd-Stage)

In the third step, iterative calculations were performed while correcting the damping ratio until the RAO obtained from the potential program became identical to that obtained from the CFD in the second step. In this step, the prerequisite should be assumed that the CFD results must be reliable and accurate values. In the iterative process, the damping ratio was modified based on the following process.

Figure 13 shows the time series of the CFD simulation results for the roll and pitch RAOs obtained from the second stage results. From these results, the mean value (radian) of the response amplitude when the motion reached a steady state, which was between 3 s and 5 s, was calculated. It was then divided by the wave amplitude under these conditions and converted into the RAO; that is, the response amplitude value per unit wave height was calculated. Table 7 summarizes these values. Finally, for the RAO calculated through CFD as an exact value, the input values for the damping ratio in the potential program were adjusted under the same conditions to determine the damping ratios for which the RAOs calculated by both methods become equal. The procedure for determining the final damping ratio is illustrated in Figure 14. Here, $R_{Pot}$ and $R_{CFD}$ refer to RAOs derived through the potential analysis and CFD simulation, respectively. For example, firstly compare $R_{Pot}$ with the initial damping ratio $\zeta^0$ to the $R_{CFD}$. If they do not match each other, the damping ratio is updated as $\zeta^1 = \zeta^0 + \omega\Delta\zeta$. Here, $\omega$ means a relaxation factor between 0 and 1, and $\Delta\zeta$ is the correction. This process will be repeated iteratively until the difference of $R_{Pot}$ and $R_{CFD}$ reaches in a small error range $\varepsilon_a$, and obtain the final $\zeta^M$. Table 8 shows the final damping ratios in each motion derived through iterative calculations.

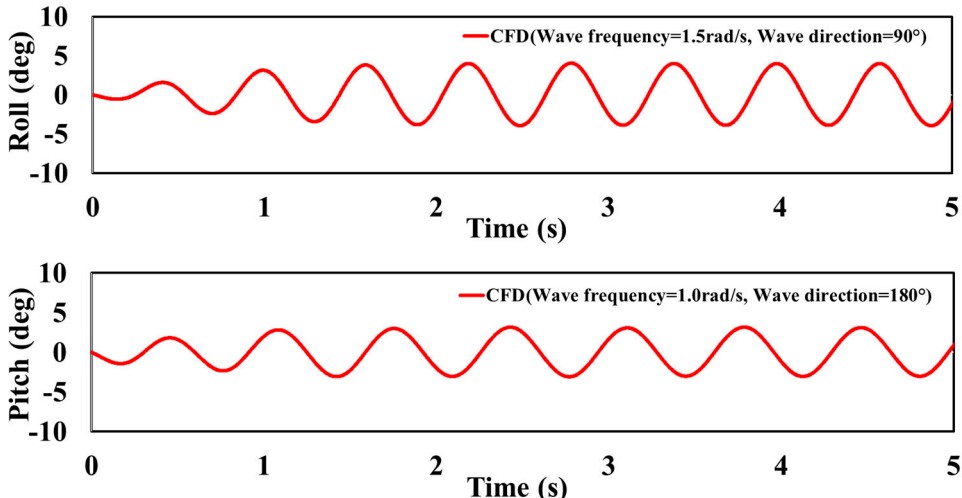

**Figure 13.** Time series of roll and pitch RAOs under tuning conditions using CFD simulations.

**Table 7.** Tuning condition between CFD and potential program.

| Motion | Wave Frequency (rad/s) | Wave Direction (°) | Time-Averaged Amplitude (rad) | RAO (rad/m) |
|--------|------------------------|---------------------|-------------------------------|-------------|
| Roll | 1.5 | 90 | 0. 0693 | 12.402 |
| Pitch | 1.0 | 180 | 0.0495 | 3.796 |

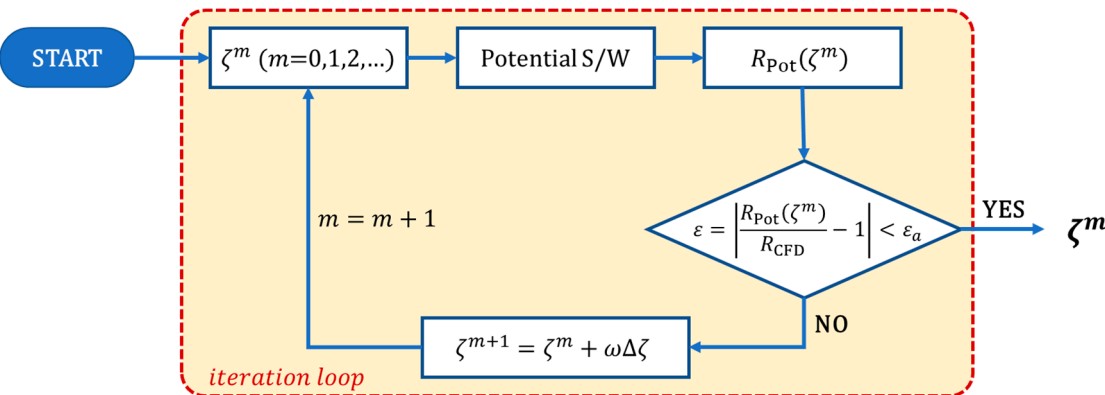

**Figure 14.** The procedure of finding the final damping ratio.

**Table 8.** Final damping ratios estimated by CMP model.

| Motion | Damping Ratio (-) |
|--------|-------------------|
| Roll | 0.112 |
| Pitch | 0.270 |

### 2.3.4. Evaluation of Seakeeping Performance (4th-Stage)

In the fourth stage, the ship response spectrum was calculated by multiplying the modified RAO calculated in the third stage with the wave spectrum in the Shinan Sea, mentioned in Section 2.2, and the seakeeping performance was evaluated for the six items listed in Appendix D.

The reference location used to calculate the RAO for each evaluation item was selected according to NORDFORSK [31] and NATO [32]. However, the ship in this study had no bridge; therefore, the reference location for lateral and vertical acceleration was substituted with the approximate bridge location. For this, we selected a location with an x-coordinate

0.25 Length between perpendiculars (LBP) away from the center of the hull and a z-coordinate 4.8 m from the keel. This location is approximately 1.5 m away from the deck of the barge; therefore, we considered it to be a reasonable location even when a person is standing. The reference location for deck wetness was the bow deck and that for slamming was the bottom of the hull, 0.15 LBP away from the bow.

## 3. Results: Validation and Discussions

### 3.1. Experiments for Validation

Experiments were conducted in an experimental wave tank (EWT) at Changwon National University, South Korea, which was 20 m long, 14 m wide, and had a water depth of 19.5 m. A 1/49 scale model ship is installed in a state that allows free 6-DOF motion by applying a soft mooring spring. After installing the tension gauges at the four angle points, the tension gauges, mooring lines, and springs were connected to the model ship. Figure 15 depicts a schematic of the model installation with sensors, soft mooring, and dimensions. An optics-based system (V120:TRIO, OptiTrack, Corvallis, OR, USA) and inertial measurement unit (IMU) sensors were used in this experiment to measure 6-DOF motion.

To conduct the seakeeping test for the model ship in regular waves, the full-scale test conditions should be scaled according to the model size. The ship speed condition at full scale is 3.086 m/s and that at the model scale is 0.441 m/s, following the Froude similarity. Regular waves with a frequency range of 0.65–1.80 rad/s at full scale are generated throughout the seakeeping test. The wave frequency in the model scale increases by the square root of the scale ratio; therefore, it corresponds to the wave frequency range of 4.55–12.60 rad/s, as listed in Table 9. The wave height condition for each frequency should be determined according to the International Towing Tank Conference (ITTC) recommendation for seakeeping experiments [33], that is, the wave height should be selected under the condition of a small wave slope (wave height/wavelength < 1/50) to obtain results analogous to the linear surface-wave theory. In addition, the model ship is symmetrical and has the same seakeeping performance for waves incident on the starboard and port sides. Thus, the experiment was conducted for a total of seven directions at a wave direction interval of 30°; waves incident on the bow are defined at 180° and on the stern at 0°.

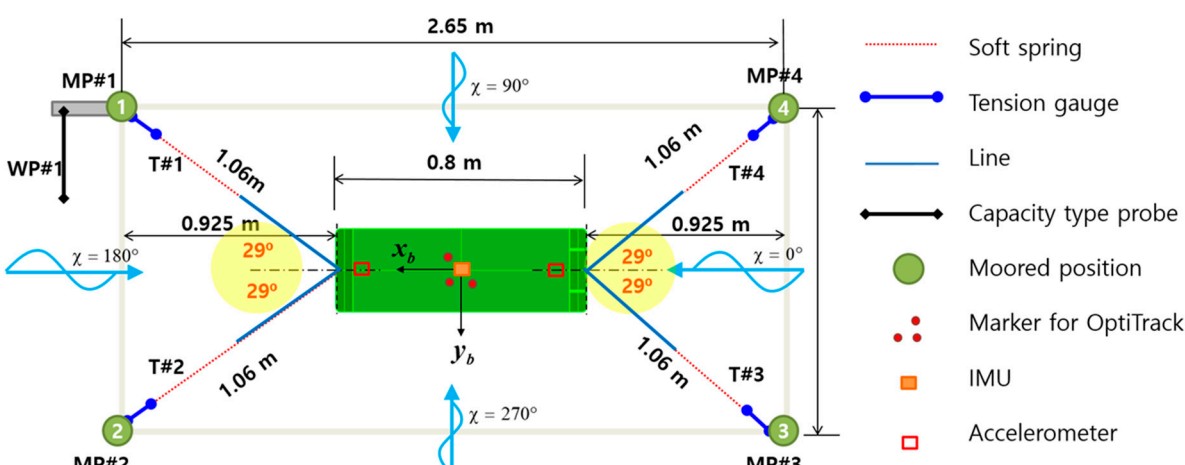

**Figure 15.** Schematic for model installation.

**Table 9.** Wave condition in model scale.

| No. | Frequency (rad/s) | Wave Length (m) | Wave Length/Ship Length | Wave Height (cm) |
|---|---|---|---|---|
| RW01 | 4.550 | 2.977 | 3.722 | 5.954 |
| RW02 | 4.900 | 2.567 | 3.209 | 5.134 |
| RW03 | 5.250 | 2.236 | 2.795 | 4.472 |
| RW04 | 5.600 | 1.965 | 2.457 | 3.930 |
| RW05 | 5.950 | 1.741 | 2.176 | 3.482 |
| RW06 | 6.650 | 1.394 | 1.742 | 2.788 |
| RW07 | 7.000 | 1.258 | 1.572 | 2.516 |
| RW08 | 7.700 | 1.040 | 1.300 | 2.080 |
| RW09 | 8.050 | 0.951 | 1.189 | 1.902 |
| RW10 | 8.400 | 0.874 | 1.092 | 1.748 |
| RW11 | 9.100 | 0.744 | 0.930 | 1.488 |
| RW12 | 10.150 | 0.598 | 0.748 | 1.196 |
| RW13 | 11.200 | 0.491 | 0.614 | 0.982 |
| RW14 | 11.900 | 0.435 | 0.544 | 0.870 |
| RW15 | 12.600 | 0.388 | 0.485 | 0.776 |

It is well known that roll damping is significantly affected by viscous effects. Therefore, the result calculated using potential theory may overestimate the roll amplitude at the natural frequency. The roll damping coefficients required to account for viscous effects can be estimated from the process summarized in Appendix C using the results of the roll decay test. Thus, to estimate roll damping, a roll decay test was carried out by forcibly the model by 8° under static conditions. The change in roll angle with time was measured using an IMU located at the center of the ballast tank, and the results are plotted in Figure 16, Figure 17 shows a plot corresponding to the analyzed results of the roll damping coefficient calculated according to the process described in Appendix C. By analyzing the roll-decay test results using the Froude energy method, the linear coefficients of the parabolic regression function provide $p_1$ and the quadratic term delivers $p_2$. For the definitions of $p_1$ and $p_2$, please see Appendix C. Roll damping coefficients and related parameters are listed in Table 10.

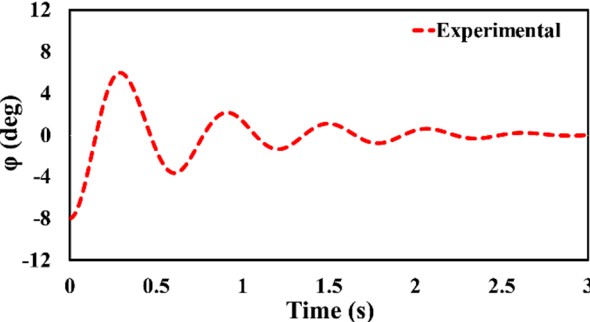

**Figure 16.** Time series of roll-decay test.

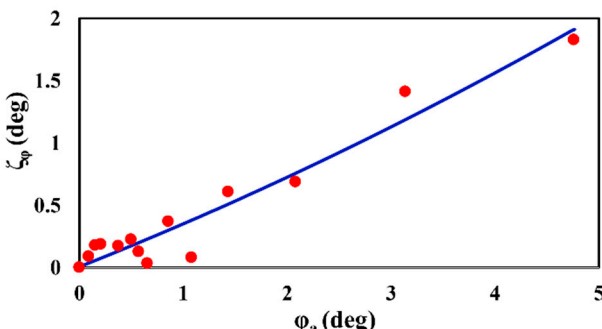

**Figure 17.** Estimation of roll-damping coefficient.

**Table 10.** Roll-damping coefficient and ratio in static condition.

| Motion | $p_1$ (s$^{-1}$) | $p_2$ (rad$^{-1}$) | $\hat{B}_{44}$ (-) | $B_{44}$ (kg·m$^2$/s) | $\zeta$(-) |
|--------|------------------|---------------------|---------------------|------------------------|-----------|
| Roll | 2.4563 | 0.0077 | 0.0587 | 8,943,703 | 0.1197 |

### 3.2. Comparison with Experiment

#### 3.2.1. Roll Decay Test in Static Condition

To verify the CFD tool currently used with the results of the experiment, a roll-decay simulation by STAR-CCM+ 15.02 was performed on the model-scale ship. Figure 18 shows a comparison of the experimental and numerical results of the roll-decay test on the model ship. For a quantitative comparison of the period and peak values, the average values of the four periods were compared with those of the experiment and the average error rate was calculated. These values are summarized in Table 11. Relative errors between the experiment and simulations were approximately 1.7% and 6.6% for the period and peak values, respectively. This implies that the results of the present CFD simulations were in good agreement with those of the experiment. The roll-damping ratio was derived through CFD using the procedure detailed in Appendix C. As shown in Table 12, the relative error between the roll-damping ratio predicted by the CFD simulations and that measured experimentally was approximately 6.6%. Hence, using CFD, a damping ratio very close to the experimental value was obtained, thereby verifying the accuracy of the CFD simulations applied in the CMP model.

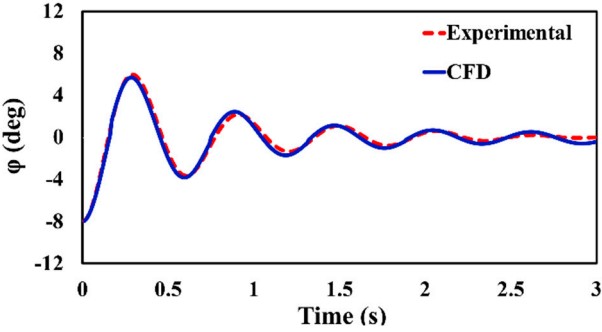

**Figure 18.** Time series of roll decay test compared with CFD simulations.

**Table 11.** Comparison of roll decay results between EWT and CFD.

| Items | Period (s) | Peak Value 1 (°) | Peak Value 2 (°) | Peak Value 3 (°) | Peak Value 4 (°) |
|-------|-----------|-------------------|-------------------|-------------------|-------------------|
| Experiment | 0.59 | 5.98 | 2.17 | 1.10 | 0.61 |
| CFD | 0.58 | 5.68 | 2.42 | 1.12 | 0.66 |
| Relative errors (%) | 1.7 | 5.0 | 11.5 | 1.8 | 8.2 |
| | | | Average 6.6 | | |

**Table 12.** Comparison of roll damping ratio between experiments and CFD in static condition.

| Items | Roll Damping Ratio, $\zeta(-)$ |
|---|---|
| Experiment | 0.1197 |
| CFD | 0.1277 |
| Relative error (%) | 6.6 |

### 3.2.2. Motion RAO in Regular Waves

The accuracy of the ship motion simulation results for regular waves calculated using the CMP model was verified by comparing them with the model test results. To examine the motion analysis results in Section 2.3.3, using the corrected final damping ratio, the potential analysis results and CMP model results for the roll and pitch RAOs were compared with the experimental data, as presented in Figures 19 and 20. As described in Section 3.1, "Experimental_OptiTrack" denotes the optics-based system, and "Experimental_IMU" denotes the motion test results for each frequency measured by the IMU sensor. In the figure, the damping ratio of the potential analysis result is indicated by a solid line for 0.05, a long-dotted line for 0.1, and a short-dotted line for 0.15. The thickest solid line represents the analysis result using the CMP method. However, the results labeled "CFD" at a 90° wave direction for roll and 180° wave direction for pitch are the results from independently conducted CFD simulations under natural frequency conditions detailed in Section 2.3.3. For the roll motion derived using the CMP model in Figure 19, the largest motion occurred in the 90° wave direction, that is, the beam-sea condition. Under this condition, the experimental results were compared with the potential analysis results that applied an arbitrary damping ratio of 0.10 and the CMP model results. The relative errors of the potential analysis for the peak period and amplitude were approximately 6.9% and 9.9%, respectively, whereas the CMP model exhibited relative errors of approximately 3.5% and 1.2%, respectively. Moreover, in the case of the pitch in Figure 20, the experiment exhibited the largest amplitude at 150°, whereas the CMP model exhibited the largest amplitude at a wave direction of 180°, that is, the head-sea condition. Under these conditions, the potential analysis results without the damping ratio had the same frequency of the peak RAO as in the experiment; however, the amplitude exhibited a large relative error of approximately 41.6%. The simulation applying the CMP model also exhibited a peak period that was highly consistent with the experimental results. However, the relative error in the amplitude was substantially reduced to approximately 5.2%, indicating that the amplitude is close to that obtained experimentally.

These results demonstrate that the motion analysis results that apply the roll and pitch damping ratios derived using the CMP model are closer to the experimental results than the potential analysis results obtained using arbitrary damping ratios. Hence, more reliable roll and pitch motion analysis results can be obtained by applying the CMP model to the motion analysis of small ships in waves.

### 3.3. Comparison of Computational Time

This section calculates the required time for motion analysis by the CMP model, compares it with those for the potential analysis and full CFD simulations, and discusses the computational efficiency of the three models. Although it is difficult to compare the models directly, accurately, and quantitatively, we make an approximate relative comparison by calculating the simulation time required by each model based on motion analysis using a linear approximation, the same computer specifications, 1-DOF, one wave heading, and 10 frequency cases.

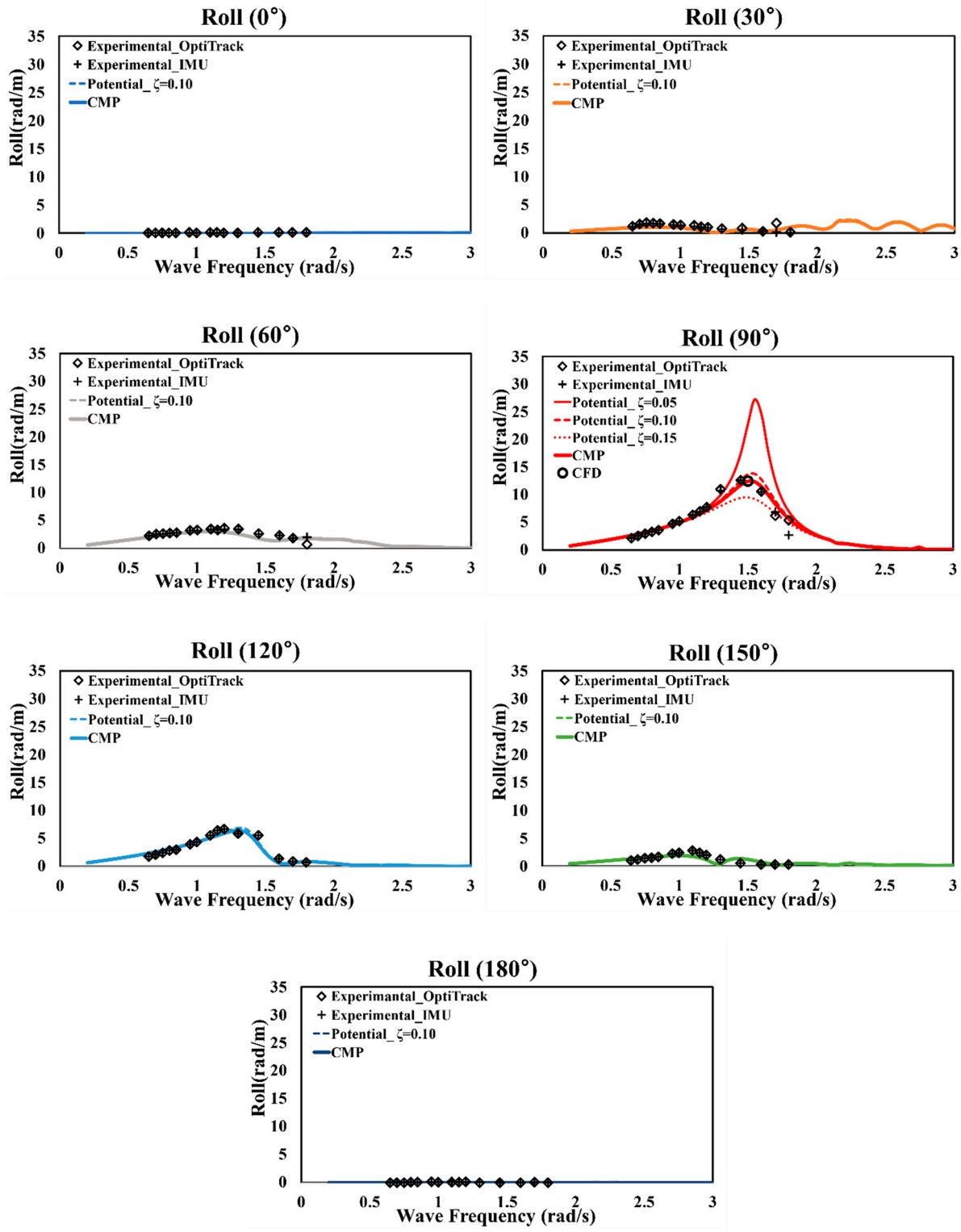

**Figure 19.** Comparison of roll RAO.

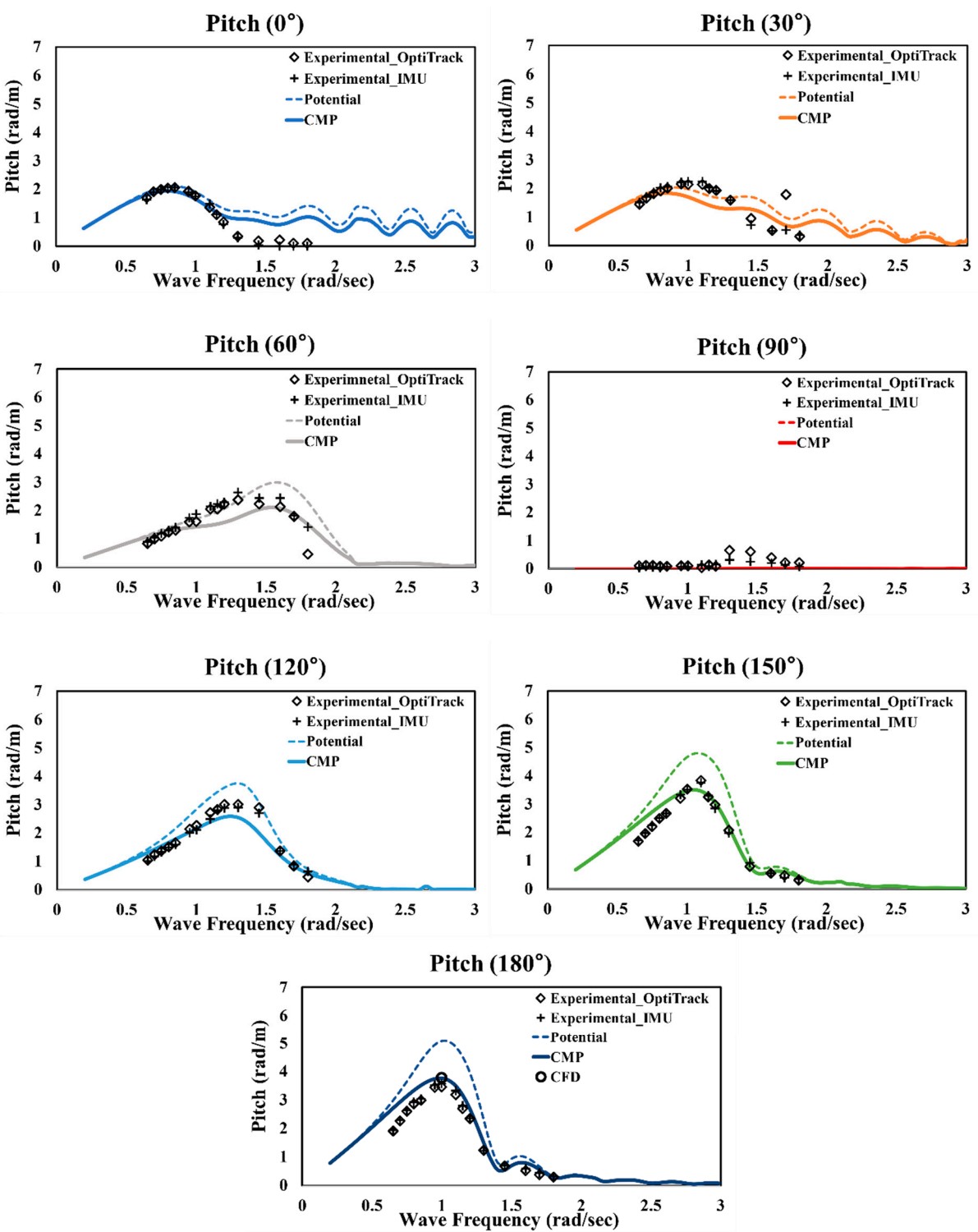

**Figure 20.** Comparison of pitch RAO.

Table 13 presents information on the analysis conditions used for each simulation and the computational time. A simulation of the potential program, WAVELOAD-FD, was performed using an Intel i7-5960X CPU @ 3.00 GHz and 8 cores with 6-DOF motion, seven wave directions, and 57 frequency domains. The total computational time for this analysis was 25,200 s. Here, if a single core is used and the analysis of one frequency and wave direction at 1-DOF is assumed to be one case, then the computational time per case would be 0.66 s. Moreover, the CFD simulations using STAR-CCM+15.02 were performed

on a 48-core X 2 Intel Xeon Platinum 9242 2.3 GHz processor with one frequency, wave direction, and 1-DOF. Here, the analysis time required per case was calculated to be 18.75 s. Motion analysis generally requires data in numerous frequency domains, including peak frequencies. However, here, the total motion analysis time estimate was based on the analysis time taken for 10 frequency cases. Accordingly, if the motion analysis is performed under conditions of 1-DOF, one wave heading, and 10 wave frequency cases using a single core, then the required time is 6.5 s for WAVELOAD-FD and 187.5 s for STAR-CCM+15.02. Additionally, the time taken for CMP analysis under the same conditions can be viewed as the sum of the required time for motion analysis of 10 frequency cases using WAVELOAD-FD and the required time for motion analysis of one case using STAR-CCM+15.02. As described earlier in the CMP model procedure, this is because the CMP model first performs an initial potential analysis in all wave directions and wave period cases at 6-DOF, and then performs CFD analysis for the wave period and wave height cases at the peak position of the RAO. Then, only the damping ratio is changed, and an additional potential analysis is performed. However, this did not considerably impact the overall computational time because the corrected result was obtained very quickly by the CMP model based on the initial potential analysis result. Table 14 compares the quantitative analysis times. The results indicate that although the CMP model takes somewhat longer than the existing potential analysis program, it produces results much faster than CFD analysis. Thus, through this comparison of calculation time for analysis, we confirmed that the proposed model is more accurate than the potential analysis and more time-efficient than the CFD simulations.

**Table 13.** Calculation of computational time according to simulation conditions.

| Case of Computational Time Calculation | WAVELOAD-FD | STAR-CCM+ |
|---|---|---|
| (a) Number of nodes | 16 | 384 |
| (b) 6-DOF cases | 6 | 1 |
| (c) Wave frequency cases | 57 | 1 |
| (d) Wave heading cases | 7 | 1 |
| (e) Total computational time (s) | 25,200 | 7200 |
| Required time for motion analysis in 1 case (s) {(e)/[(a) × (b) × (c) × (d)]} | 0.66 | (f) 18.75 |
| Required time for motion analysis in 10 frequency cases (s) | (g) 6.60 | 187.50 |

**Table 14.** Comparison of computational time for motion analysis in 10 frequency cases.

| Numerical Model | WAVELOAD-FD |
|---|---|
| WAVELOAD-FD | 6.60 |
| STAR-CCM+ | 187.50 |
| CMP | 25.35 [(f) + (g)] |

*3.4. Comparison of Seakeeping Performance*

Tables 15 and 16 present an evaluation of seakeeping performance results. To evaluate the seakeeping performance for the roll and pitch, we compared the potential analysis results that apply an arbitrary damping ratio of 0.05, CMP model analysis results, and model experiment results. The experimental values for vertical and lateral acceleration, deck wetness, or slamming were not available; therefore, only the potential analysis and CMP model results were compared to investigate the differences.

**Table 15.** Evaluation of seakeeping performance on roll and pitch motions.

| Motion Response | | 0° | 30° | 60° | 90° | 120° | 150° | 180° | Criterion |
|---|---|---|---|---|---|---|---|---|---|
| Roll (°) | Potential | 0.00 | 0.34 | 0.63 | 3.18 | 1.52 | 0.33 | 0.00 | |
| | CMP | 0.00 | 0.21 | 0.52 | 2.03 | 1.01 | 0.25 | 0.00 | 8.0 |
| | Exp. | 0.00 | 0.18 | 0.67 | 1.76 | 0.75 | 0.19 | 0.00 | |
| Pitch (°) | Potential | 0.32 | 0.37 | 0.56 | 0.00 | 0.65 | 0.67 | 0.62 | |
| | CMP | 0.25 | 0.29 | 0.40 | 0.00 | 0.46 | 0.47 | 0.44 | 3.0 |
| | Exp. | 0.37 | 0.43 | 0.64 | 0.41 | 0.70 | 0.56 | 0.49 | |

**Table 16.** Evaluation of seakeeping performance on vertical and lateral accelerations, deck wetness, and slamming.

| Motion Response | | 0° | 30° | 60° | 90° | 120° | 150° | 180° | Criterion |
|---|---|---|---|---|---|---|---|---|---|
| Vertical acceleration (g) | Potential | 0.00 | 0.00 | 0.01 | 0.02 | 0.02 | 0.02 | 0.01 | 0.4 |
| | CMP | 0.00 | 0.00 | 0.01 | 0.02 | 0.01 | 0.01 | 0.01 | |
| Lateral acceleration (g) | Potential | 0.00 | 0.01 | 0.02 | 0.02 | 0.03 | 0.02 | 0.00 | 0.2 |
| | CMP | 0.00 | 0.01 | 0.02 | 0.01 | 0.03 | 0.02 | 0.00 | |
| Deck wetness (No./hour) | Potential | 0.00 | 0.00 | 0.00 | 0.00 | 0.00 | 0.00 | 0.00 | 30 |
| | CMP | 0.00 | 0.00 | 0.00 | 0.00 | 0.00 | 0.00 | 0.00 | |
| Slamming (No./hour) | Potential | 0.00 | 0.00 | 0.00 | 0.00 | 0.00 | 0.00 | 0.00 | 20 |
| | CMP | 0.00 | 0.00 | 0.00 | 0.00 | 0.00 | 0.00 | 0.00 | |

Table 15 presents the evaluation of the seakeeping performance results for the roll and pitch, and Figure 21 shows radial graphs of these results. Seakeeping performance analysis in this study was conducted from 0° to 180°. However, as the target barge is symmetrical about the xz-plane, the results in the wave directions from 180° to 360° are displayed symmetrically. The potential analysis overpredicted the SSA by over 80% compared to the experimental results for a roll with a wave-direction condition of 90°, where motion is the most severe. However, the CMP model under the same conditions yielded a value that was considerably closer to the experimental result than the potential result, with an error of approximately 15% relative to the experimental result. The CMP model results were closer to the experimental results for the other wave directions, except for the quartering sea (wave direction of 60°). Moreover, for a pitch under a wave direction condition of 180°, where motion is the most severe, the SSA calculated using the potential theory had an error of approximately 27% compared with the experimental results. The CMP model exhibits an error of approximately 10%. For other wave direction conditions, the potential analysis results were closer to the experimental results. This error likely occurred because the modified damping ratio for the 180° wave direction condition, where the peak RAO occurred, was applied equally to other wave directions when applying the CMP model. Additionally, for the 90° wave direction, the experimentally obtained value was 0.41°, whereas that obtained from the simulations was close to 0°. In the experiment, as speed increased, the difference in wave elevation at the front and rear of the hull increased, resulting in a pressure difference. Hence, when the ship velocity and wave direction are at 90° with respect to each other, trim occurs owing to the difference in the fluid force acting on the stern and bow, which causes an increase in pitch motion. This was observed in the CFD results shown in Figure 22. In contrast, the potential theory, by nature, does not reflect the effect of fluid viscosity; therefore, the pitch at the 90° wave direction, converted to the seakeeping performance units for evaluation, was close to 0. This is judged to be a limitation of the potential theory-based analysis and should be addressed in future research.

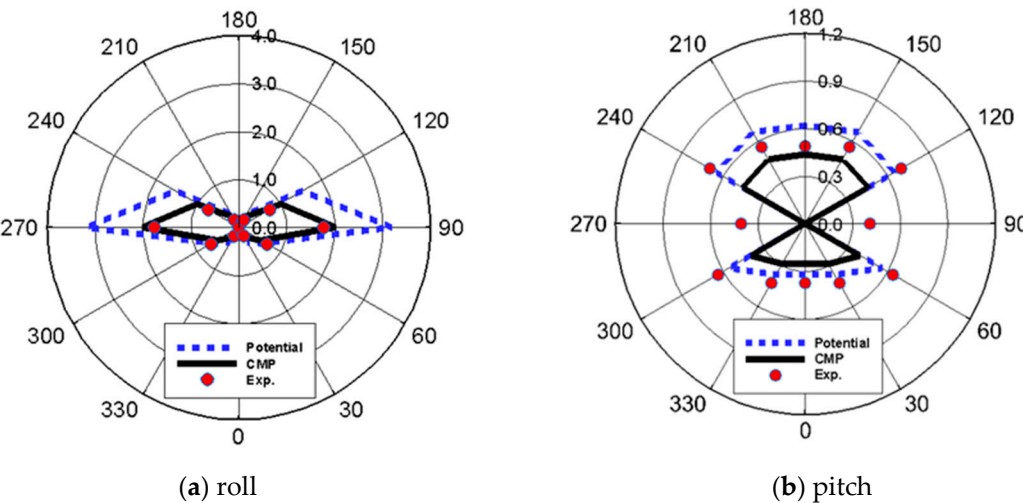

(**a**) roll                                                                                          (**b**) pitch

**Figure 21.** Azimuth diagram of SSA.

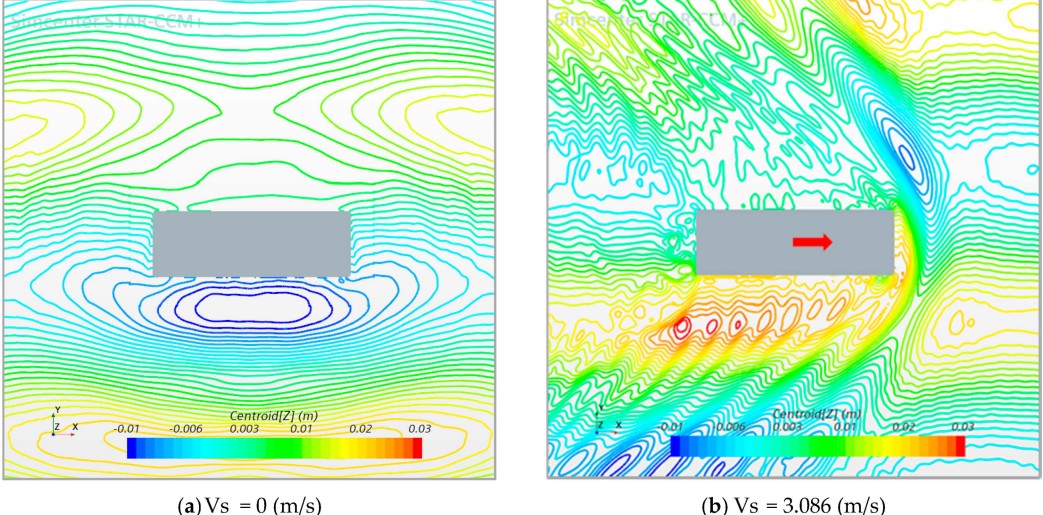

(**a**) Vs = 0 (m/s)                                                                          (**b**) Vs = 3.086 (m/s)

**Figure 22.** Wave contour in wave direction of 90° at different ship speeds.

Table 16 shows the evaluation of the seakeeping performance results for the other four evaluation metrics (vertical and lateral acceleration, deck wetness, and slamming). Because there were no experimental results, we compared only the potential analysis and CPM model results. Both sets of results satisfied the seakeeping performance criteria for all wave directions, and the differences between the results of both methods were not substantial. This may be because the state of the target sea area was very low. It is necessary to evaluate seakeeping performance in more severe sea states in future studies.

## 4. Conclusions

The damping ratio is arbitrarily applied based on experience when performing a potential program analysis to analyze the seakeeping performance of ships in shipbuilding-related research. Therefore, with reference to the evaluation of the seakeeping performance procedure proposed by Kim et al. [15], we propose the CMP model in this study, which is a new motion analysis method that addresses the shortcomings of existing motion analysis programs in inversely predicting the damping ratio. Motion analysis was performed by applying the damping ratio predicted by the CMP model, based on which seakeeping performance analysis was performed. In addition, a model experiment was conducted to ensure the reliability of the results derived using the proposed model. The following conclusions can be drawn from the comparison of the analysis results:

- CFD reliability from roll decay test: The damping ratio derived from the free roll decay model test was 0.1197, and the roll damping ratio derived from CFD was 0.1277, which exhibited a relative error of approximately 6.6%. This CFD simulation result was very close to the experimental value, thereby verifying the accuracy and reliability of the CFD used in the CMP model process.
- RAO from motion analysis: For roll motion, the motion analysis results were examined under beam sea conditions. When the damping ratio derived from the CMP model was applied, the peak period and amplitude exhibited errors of approximately 3.5% and 1.2%, respectively, compared to the experiment. For pitch motion, the motion analysis results under head-sea conditions were examined. When the damping ratio derived using the CMP model was applied, the peak period measured in the experiment was consistent with the simulation results, and the amplitude exhibited an error of only 5.2%. This indicates that results comparable to those of the model experiment can be obtained using a motion analysis that applies an appropriate damping ratio. This has typically been applied based on experience and estimated using the CMP model for potential programs. These findings suggest that motion analysis using the CMP model yields more reliable results.
- Effects on computational time: The CMP model requires approximately 3.8 times more computational time than that required for the potential analysis. However, under certain conditions, the CMP model produces results that are close to experimental results. Further, the model can derive results 7.4 times faster than CFD simulations, and the error when compared with the experimental results was approximately 5%. This indicates that the CMP model can derive more accurate results than those derived by the potential analysis and takes less time than required for CFD.
- Evaluation of seakeeping performance: For pitch and roll, the results of the potential analysis, CMP-based analysis, and experiment were compared. Under wave direction conditions, where each motion was the largest, the CMP model yielded seakeeping performance results that were close to the experiment. However, for several following and quartering sea conditions, the potential analysis results were closer to the experimental results than those of the CMP model. This error in the CMP model occurred because the damping ratio was modified only for the wave direction condition where motion was the largest, but it was applied to other wave directions as well. The evaluation results of the potential analysis and CMP model for vertical and lateral acceleration, deck wetness, and slamming did not differ significantly, which can be attributed to the very low sea state of the target sea area. Therefore, it is necessary to compare, examine, and evaluate the results across different sea states in future research.

**Author Contributions:** Conceptualization J.-C.P.; methodology, J.-C.P. and J.-B.P.; experiment, H.-K.Y.; software, S.N.; validation, S.N.; formal analysis, S.N.; investigation, S.N.; resources, S.N. and H.-K.Y.; data curation, S.N.; writing—original draft preparation, S.N.; writing—review and editing, J.-C.P.; visualization, S.N.; supervision, J.-C.P.; project administration, J.-C.P. All authors have read and agreed to the published version of the manuscript.

**Funding:** This research was a part of the project titled "Development of coastal garbage collection technology in difficult-to-access areas", funded by the Ministry of Oceans and Fisheries, South Korea.

**Data Availability Statement:** Not applicable.

**Conflicts of Interest:** The authors declare no conflict of interest.

## Appendix A

The TMA spectrum reflects the influence of a finite water depth by multiplying the JONSWAP spectrum by the shape function developed by Kitaigordskii et al. [34]. For the JONSWAP spectrum, the form proposed by Goda [35] was used, and for Kitaigordskii's

shape function, the approximate formula proposed by Thompson and Vincent [36] was used. Equation (A1) represents the TMA spectrum:

$$S(\omega) = \frac{\alpha H_s^2 \omega_p^4}{\omega} \exp\left[-1.25\left(\frac{\omega_p}{\omega}\right)^4\right] \gamma^{exp[-\frac{(\omega-\omega_p)^2}{2\sigma^2\omega_p^2}]} \varphi(\omega_h) \tag{A1}$$

where the Philips parameter $\alpha$ can be calculated by Equation (A2), which depends on the peak enhancement factor $\gamma$, defined in Equation (A3), according to DNV-OS-E301 [37]. Additionally, $\omega$ and $\omega_p$ are the wave frequency and peak frequency, respectively, and $\sigma$ is a variable parameter defined according to the relationship between $\omega$ and $\omega_p$, as given in Equation (A4). The transformation factor $\varphi(\omega_h)$ represents the effect of the water depth $h$, and it can be determined by Equation (A5). Here, $\omega_h$ can be calculated by Equation (A6). The spectral peak period $T_p$ is estimated by Equation (A7).

$$\alpha = \frac{0.0624(1.094 - 0.01915\ln\gamma)}{0.23 + 0.0336\gamma - \frac{0.185}{1.9+\gamma}} \tag{A2}$$

$$\gamma = \begin{cases} 5, \ if \ \left(\frac{T_p}{\sqrt{H_s}} \leq 3.6\right) \\ e^{5.75-1.15\frac{T_p}{\sqrt{H_s}}}, \ if \ \left(3.6 \leq \frac{T_p}{\sqrt{H_s}} \leq 5\right) \\ 1, \ if \ \left(5 \leq \frac{T_p}{\sqrt{H_s}}\right) \end{cases} \tag{A3}$$

$$\sigma = \begin{cases} 0.07, \ if \ (\omega \leq \omega_p) \\ 0.09, \ if \ (\omega > \omega_p) \end{cases} \tag{A4}$$

$$\varphi(\omega_h) = \begin{cases} 0.5\omega_h^2, \ if \ (\omega_h \leq 1) \\ 1 - 0.5(2 - \omega_h)^2, \ if \ (1 < \omega_h \leq 2) \\ 1, \ if \ (\omega_h > 2) \end{cases} \tag{A5}$$

$$\omega_h = \omega\sqrt{\frac{h}{g}} \tag{A6}$$

where $g$ is gravitational acceleration.

$$T_P = \frac{T_z}{\sqrt{\frac{5+\gamma}{10.89+\gamma}}} \tag{A7}$$

where $T_z$ is the mean zero-crossing wave period.

**Appendix B**

For the GCI theory, the Richardson extrapolation method was used according to the mesh density, which is a method based on Taylor expansion to obtain higher-order estimates of continuum values from discrete values [30,38]. First, this is discretized with respect to function $f$ to obtain the series in Equation (A8). Here, $h$ is the grid spacing, and $f_n$ is the nth-order derivative term.

$$f = f_{h=0} + f_1 h + f_2 h^2 + f_3 h^3 + \cdots \tag{A8}$$

We assume that $f_1 = 0$ in the above equation; then, $f$ is considered to be a second-order term, and $F_0$ is the value when the grid spacing is 0. Under the second-order assumption, by calculating the fine-grid solution $f_1$ and coarse-grid solution $f_2$ in the fine grid spacing $h_1$ and coarse grid spacing $h_2$, respectively, and then combining the two resulting equations, we obtain Equation (A9). Here, $r$ is the refinement factor between the coarse and fine grids, $r = h_2/h_1$, and $p$ is the formal order of the accuracy of the algorithm. Here, if the grid

refinement factor $r$ is considered to be 2 and is expressed as a quadratic equation, we obtain Equation (A10), which is a generalized Richardson extrapolation.

$$f_{h=0} \cong f_1 + \frac{f_1 - f_2}{r^p - 1} \tag{A9}$$

$$f_{h=0} \cong \frac{4}{3}f_1 - \frac{1}{3}f_2 \tag{A10}$$

The fine-grid Richardson error estimator approximates the error in $f_1$ by comparing it with $f_2$, as defined in Equation (A11). Moreover, the coarse-grid Richardson error estimator approximates the error in $f_2$ by comparing it with $f_1$, as defined in Equation (A12). Here, $\varepsilon = f_2 - f_1$, which indicates the difference in the grid level.

$$E_1^{fine} = \frac{\varepsilon}{1 - r^p} \tag{A11}$$

$$E_2^{coarse} = \frac{r^p \varepsilon}{1 - r^p} \tag{A12}$$

The GCI is a percentage of the computed value or asymptotic solution, and it indicates the error associated with how far the solution is from the asymptotic value. A small GCI value indicates that the computation was within the asymptotic range. Finally, the GCIs of the fine and coarse grids were defined as in Equations (A13) and (A14), respectively:

$$GCI_1^{fine} = F_s|E_1| \tag{A13}$$

$$GCI_2^{coarse} = F_s|E_2| \tag{A14}$$

where $F_S$ is the safety factor, $F_S = 3$ is recommended for two grids, and $F_S = 1.25$ for three or more grids.

**Appendix C**

The procedure used to estimate the damping coefficient was described by Rodríguez et al. [39]. First, the roll decay motion of a ship in calm water can be expressed as a 1-DOF equation, given by Equation (A15), where $\varphi(t)$ denotes the instantaneous roll motion, $I_{44}$ is the mass moment of inertia, and $a_{44}$ and $c_{44}$ represent the added mass moment of inertia and hydrostatic restoring coefficient, respectively. $b_{44}(\varphi)$ is the damping coefficient, which depends on the roll amplitude. Equation (A15) can be rewritten as Equation (A16). Here, $\omega_n^2$ and $p(\varphi)$ can be calculated using Equations (A17) and (A18), respectively:

$$(I_{44} + a_{44})\ddot{\varphi}(t) + b_{44}(\varphi)\left(\dot{\varphi}(t)\right) + c_{44}\varphi(t) = 0 \tag{A15}$$

$$\ddot{\varphi} + p(\varphi)\dot{\varphi} + \omega_n^2\varphi = 0 \tag{A16}$$

$$\omega_n^2 = \frac{mgGM_T}{I_{44} + a_{44}} \tag{A17}$$

$$p(\varphi) = \frac{b_{44}(\varphi)}{I_{44} + a_{44}} \tag{A18}$$

The damping moment can be expressed in terms of the linear and quadratic contributions, as shown in Equation (A19). Assuming a harmonic roll motion and equivalent dissipated energy for both damping representations, an additional relationship between the coefficients $p$, $p_1$, and $p_2$ can be established, as shown in Equation (A20).

$$p(\varphi)\dot{\varphi} = p_1\dot{\varphi} + p_2\dot{\varphi}|\dot{\varphi}| \tag{A19}$$

$$p(\varphi_a) = p_1 + p_2\frac{16}{3T_k}\varphi_a \tag{A20}$$

where the roll amplitude is $\varphi_a = (\varphi_k + \varphi_{k+1})/2$, $\varphi_k$ and $\varphi_{k+1}$ denote two successive peaks in the roll decay motion, and $T_k$ is the roll period. Roll damping coefficients p, $p_1$, and $p_2$ were obtained from roll decay time records. In this study, the Froude energy method was used to analyze roll decay time records. This approach is based on the energy loss balance in each half cycle. The energy dissipated by the damping term is equal to the variation in the potential energy (work done by the restoring moment) when the kinetic energies at the initial and final positions are zero. Assuming a linear plus quadratic damping form and linear restoring moment in the roll decay equation, the energy balance is given by Equation (A21). This can be expressed by Equation (A22), with $d\varphi = \varphi'dt$.

$$\int_{\varphi_k}^{\varphi_{k+1}}\left[\varphi' + p_1\varphi' + p_2\varphi'|\varphi'|\right]d\varphi = \int_{\varphi_k}^{\varphi_{k+1}}\omega_n^2\varphi d\varphi \tag{A21}$$

$$\int_0^{T_k/2}\left[\varphi' + p_1\varphi' + p_2\varphi'\left|\varphi'\right|\right]\varphi'dt = \int_{\varphi_k}^{\varphi_{k+1}}\omega_n^2\varphi d\varphi \tag{A22}$$

The integration of each term gives:

$$\int_0^{T_k/2}\varphi'(t)\varphi'(t)dt = 0,$$

$$\int_0^{T_k/2}p_1\varphi'(t)\varphi'(t)dt = p_1\frac{\pi^2}{T_k}\varphi_a^2, \tag{A23}$$

$$\int_0^{T_k/2}p_2\varphi'(t)\left|\varphi'(t)\right|\varphi'(t)dt = p_2\frac{16\pi^2}{3T_k^2}\varphi_a^3$$

where $\varphi_a = \frac{(\varphi_k+\varphi_{k+1})}{2}$, and $\omega_n = \frac{2\pi}{T_k}$.

$$p_1\frac{\pi^2}{T_k}\varphi_a^2 + p_2\frac{16\pi^2}{3T_k^2}\varphi_a^3 = \omega_n^2(\varphi_{k+1} - \varphi_k)\varphi_a \tag{A24}$$

Denoting $\delta\varphi = (\varphi_k - \varphi_{k+1})$

$$\delta_\varphi = p_1\frac{T_k}{4}\varphi_a + p_2\frac{4}{3}\varphi_a^2 \tag{A25}$$

Equation (A25) represents a quadratic function of $\delta\varphi$ against $\varphi_a$, where $p_1$ and $p_2$ can be obtained using a regression procedure. In this approach, k is assumed to be a positive peak, whereas the successive $k+1$ peak is a negative peak. The non-dimensionalized roll damping coefficient is expressed in Equation (A26), and the equation for calculating the damping ratio for the critical damping coefficient is expressed in Equation (A27).

$$\hat{B}_{44} = \frac{B_{44}}{\rho\nabla B^2}\sqrt{\frac{B}{2g}} \tag{A26}$$

$$\zeta = \frac{B_{44}}{2(I_{44} + A_{44})\omega_n} \tag{A27}$$

**Appendix D**

The seakeeping performance was evaluated for the six items as listed in Table A1. The evaluation items and reference location used to calculate the RAO for each evaluation item was selected according to NORDFORSK [31] and NATO [32]. As shown in Equation (A28), a quantitative evaluation for each item was performed using the ship response spectrum

by applying the convolution of the RAO calculated at each reference location and the wave spectrum of the target sea area.

$$S_\alpha(\omega) = S_\omega(\omega) \times |RAO(\omega)|^2 \tag{A28}$$

where $S_\alpha(\omega)$ denotes the ship response spectrum, $S_\omega(\omega)$ denotes the wave spectrum, and $\omega$ denotes the wave frequency.

The response results for each item were converted into units for quantitative evaluation. First, the $SSA$ is calculated for roll, pitch, and vertical and lateral accelerations, which are obtained from the 0th moment $m_0$ of the response spectrum, as in Equation (A29). However, $m_0$ was calculated using Equation (A30). Roll and pitch motions were evaluated in units of $SSA$ (degrees), and vertical and lateral accelerations were expressed in units of $SSA$ (g) as the ratio to gravitational acceleration of 9.8 m/s$^2$.

$$SSA = 2\sqrt{m_0} \tag{A29}$$

$$m_n = \int_0^\infty \omega_e^n S_\alpha(\omega)\delta\omega \tag{A30}$$

where $\omega_e$ is the encounter frequency of the ship and $m_n$ is the $n$th moment of the response spectrum.

Deck wetness and slamming were evaluated according to the number of occurrences per hour obtained from the occurrence probability. The probability of deck-wetness occurrence $P_{DW}$ is the probability that the wave height will exceed the vertical displacement at the top of the bow, which is expressed in Equation (A31). The number of deck-wetness occurrences per hour $N_W$ is $P_{DW}$ divided by the mean wave period $T_z$, which is expressed in Equation (A32).

$$P_{DW} = e^{-(\frac{f_{DW}^2}{2m_{0s}})} \tag{A31}$$

$$N_W = \frac{P_{DW}}{T_z} \times 3600 = \frac{3600}{2\pi}\sqrt{\frac{m_{2s}}{m_{0s}}}e^{-(\frac{f_{DW}^2}{2m_{0s}})} \tag{A32}$$

where $f_{DW}$ is the effective freeboard considering the target bow motion of the wave, and $m_{0s}$ and $m_{2s}$ are the response spectrum moments of the relative vertical displacement motion and relative vertical speed motion of the hull at the deck wetness reference location, respectively. These values were obtained using Equation (A30) [40,41].

Slamming is the probability $P_{SLAM}$ that the speed of the bottom relative to that of the wave exceeds the speed limit when the bottom collides with the water surface and is calculated as follows:

$$P_{SLAM} = e^{-(\frac{f_{SLAM}^2}{2m_{0s}} + \frac{v_0^2}{2m_{0s}})} \tag{A33}$$

where $v_0$ is the vertical speed limit of the hull bottom relative to the wave, which can be obtained as follows:

$$v_0 = 0.093\sqrt{gL} \tag{A34}$$

where $g$ is gravitational acceleration, and $L$ is the length between perpendiculars.

The number of slamming occurrences per hour $N_S$ is $P_{SLAM}$ divided by the mean wave period $T_z$, which is expressed as follows:

$$N_S = \frac{P_{SLAM}}{T_z} \times 3600 = \frac{3600}{2\pi}\sqrt{\frac{m_{2s}}{m_{0s}}}e^{-(\frac{f_{SLAM}^2}{2m_{0s}} + \frac{v_0^2}{2m_{0s}})} \tag{A35}$$

where $f_{SLAM}$ is the vertical displacement from the slamming reference location to the draft, and $m_{0s}$ and $m_{2s}$ are the response spectrum moments to the relative vertical displacement motion and relative vertical speed motion of the hull at the slamming reference location, respectively.

**Table A1.** Seakeeping performance criteria.

| Motion Response | Reference Location | Units | Criterion |
|---|---|---|---|
| Roll | COG | *SSA* (deg) | 8.0 |
| Pitch | COG | *SSA* (deg) | 3.0 |
| Vertical acceleration | Center of bridge | *SSA* (g) | 0.4 |
| Lateral acceleration | Center of bridge | *SSA* (g) | 0.2 |
| Deck wetness | FP | No./hour | 30 |
| Slamming | 0.15 LBP abaft FP | No./hour | 20 |

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
