# Peer review of "CFD-Modified Potential Simulation on Seakeeping Performance of a Barge"

_water, doi:10.3390/w14203271_

Round 1

Reviewer 1 Report

CFD and potential calculations were performed for a barge. The contents are valuable. However, they need some improvements. So, please, consider these major review comments.

1.              We do not need specific numbers from the results in the abstract part. 

2.              It is recommended to include some references to support your statement in lines 38 - 41. "Compared to large ships, the stability of small ships 39 is at a higher risk of being affected by a wave environment, not only by  the roll motion but also by other planar motions, including the pitch  motion."

3.              Some studies that discuss the limitation of the linear theory in actual seakeeping conditions should be cited to support your statement in lines 53 to 55. "However, 54 applying linear theory to model the actual physical phenomenon may be 55 an oversimplification."

4.              English can be improved for lines 87 - 95. "However, it has limitations in predicting roll motion. This is caused by the nonlinear effects of roll damping owing to the fluid viscosity, which is not considered in potential theory. To compensate for this, the concept of artificial viscous damping was introduced into the potential theory in previous studies, thereby including viscous effects in the calculation. However, it is difficult to determine a suitable magnitude of artificial viscous damping owing to the strong nonlinearity of roll motion (Jung et al., 2005). "

5.              You should reference the statement in lines 195 and 196. "According to the sea state code classification criteria of the World Meteorological Organization."

6.              Table 2 is not a good representation because it is hard to see the distribution. You can replace it with other models, such as a 3-D bar chart.

7. Figures 7 and 8 can be improved by setting the y-axis range closer to the value range and putting legends vertically to the top right.

8.              The header "Principal" in Table 4 is obscure. As a result, elements in the "Items" column are not related to each other.

9.              In section 2.3.3, It is necessary to describe how you determined/modified the damping ratio, at least in text, better accompanied by math equations.

10. Tables  2, 3, 5, and 12 are separated into two parts by page breaks.

11.           To let the readers understand the paragraph easily, it is recommended to include equations (B.4) to (B.7) in the paragraph, starting with line 325 and ending with line 344.

12.           In Table 6, the column header "Mesh level" must be changed to one that represents the vertical set of items (i.e. {r, p, ??????? ...} ).

13.           English can be improved for lines 371 - 375.

14.           Figure 13 is not correctly inserted.

15.           In Figure 13 and Figure, step "RAOPotential" has a  branching, but no condition for it is specified.

16.           Inconsistency in spelling can be seen in the "n-th stage." For example, "the fourth and final stage" is used in line 404, while "the 3rd-stage" in line 405 and "4th-stage" in line 403 is found in the same paragraph. These expressions must be consistent for the whole article level.

17.           Table 13 can be improved by adding a middle horizontal line between the 1st and 2nd rows.

18.           In Tables 14 and 15, the column header "Program name" should be changed. See comment (12).

19.           The axis range should be decreased in Figure 20 so readers can understand the agreement or discrepancy among the methods.

20.           The statement in lines 618 and 619, "The target barge is symmetrical about the x- y axis," contradicts Figure 2. Did you mean symmetry about the xz-plane?

21.           In section 2.1,  CFD is described as used to analyze a floating object's response under waves. However, in section 3.1, the free decay test results of the roll are compared with the experiment to validate the CFD results. Why are free decay test results compared to validate the performance of CFD for predicting response in waves?

22.           In Appendix B, some of the text's math equations are disabled. For example, in lines 863 and 864, "fand coarse-grid solution fin the fine grid spacing hand coarse grid spacing h2, respectively, and then combining the two resulting equations," 

23.           Since you used multiple methods, you should mention which figure comes from which in titles and legends, especially for figures showing RAOs and time series.

24.           The Resolution of the figures is too small to read the legends and labels.

25.           Equations should be centered.

26.           Extensive editing of English language and style required.

Author Response

Please, read the attached file

Reviewer 2 Report

Please, read the attached file

Author Response

Please, read the attached file

Round 2

Reviewer 1 Report

The authors have worked on improving the draft and replied correctly to all my comments. Therefore, at this stage, I recommend the article be published.